# CMV-Specific Immune Response—New Patients, New Insight: Central Role of Specific IgG during Infancy and Long-Lasting Immune Deficiency after Allogenic Stem Cell Transplantation

**DOI:** 10.3390/ijms20020271

**Published:** 2019-01-11

**Authors:** Przemyslaw Zdziarski

**Affiliations:** 1Department of Clinical Immunology and Hospital Infection Control Team, Lower Silesian Center for Cellular Transplantation, 53-439 Wrocław, Poland; zdziarski@oil.org.pl; 2Military Institute WITI Wroclaw; 136 Obornicka Str., 50-961 Wrocław, Poland

**Keywords:** cytomegalovirus (CMV), epidemiology, host-virus interaction, CMV-specific IgG, protective IgG level, adoptive/acquired, innate immune response, hematopoietic stem cell transplantation (HSCT), secondary immunodeficiency, pentamer, β_2_-microglobulin, QuantiFERON, epitope engineering, natural killer (NK) cells

## Abstract

Although the existing paradigm states that cytomegalovirus (CMV) reactivation is under the control of the cellular immune response, the role of humoral and innate counterparts are underestimated. The study analyzed the host–virus interaction i.e., CMV-immune response evolution during infection in three different clinical situations: (1) immunodeficient CMV-positive human leukocyte antigen (HLA)-matched bone marrow recipients after immunoablative conditioning as well as immunocompetent, (2) adult, and (3) infant with primary immune response. In the first situation, a fast and significant decrease of specific immunity was observed but reconstitution of marrow-derived B and natural killer (NK) cells was observed prior to thymic origin of T cells. The lowest CMV-IgG (93.2 RU/mL) was found just before CMV viremia. It is noteworthy that the sole and exclusive factor of CMV-specific immune response is a residual recipient antibody class IgG. The CMV-quantiferon increase was detected later, but in the first phase, phytohemagglutinin (PHA)-induced IFN-γ release was significantly lower than that of CMV-induced (“indeterminate” results). It corresponds with the increase of NK cells at the top of lymphocyte reconstitution and undetected CMV-specific CD8 cells using a pentamer technique. In immunocompetent adult (CMV-negative donor), the cellular and humoral immune response increased in a parallel manner, but symptoms of CMV mononucleosis persisted until the increase of specific IgG. During infancy, the decrease of the maternal CMV-IgG level to 89.08 RU/mL followed by clinical sequel, i.e., CMV replication, were described. My observations shed light on a unique host-CMV interaction and CMV-IgG role: they indicate that its significant decrease predicts CMV replication. Before primary cellular immune response development, the high level of residual CMV-IgG (about >100 R/mL) from mother or recipient prevents virus reactivation. The innate immune response and NK-dependent IFN-secretion should be further investigated.

## 1. Introduction

Although cytomegalovirus (CMV) is the most common cause of host–virus interaction (named here host-CMV balance), it appears to be an important problem in the clinical practice of infectious disease immunology. It may be strictly local, e.g., CMV retinitis, hearing loss [1], mental retardation, cognitive defects, and neurodevelopmental abnormalities [2]. A multiorgan or systemic infectious process is observed in patients undergoing bone marrow transplantation with a high degree of morbidity and mortality. On the contrary, CMV usually produces mild or asymptomatic infections in immunocompetent individuals. The distinguishing symptoms are not observed, common symptoms are headaches, myalgia (53%), profuse sweat (50%), abdominal pain, diarrhea, recent loss of weight, dry cough (51%), and splenomegaly in 24% of the cases [3]. Clinical manifestations include also: suppression of myelopoiesis, mononucleosis-like syndrome, hepatosplenomegaly, lymphadenopathy, thrombocytopenia, hemolytic anemia, and oral mucositis. CMV mononucleosis with prolonged fever, weight loss, and lymphocytosis were also observed in immunocompetent patients [4]. Unique conditions of the affected patients and diversity of clinical manifestations of cytomegalovirus infection do not allow formation of real comparable cohort of patients without distinguishing different clinical host–virus constellation for example presented in table in discussion section (see below). This might be due to the presence of several confounding factors (e.g., CMV-antigen types, immunoparameters, immunosuppressive therapy, and immunogenetic background with inter-individual variation presented in Appendix A). It may bias the final results in multicenter U.S. National Marrow Donor Program (NMDP) and European Group for Blood and Marrow Transplantation (EBMT) studies [5,6]. Differential diagnosis and classification of CD8 T cell infiltration were quite difficult in histological examination depending on whether it was Graft-versus-host (GvH)- or virus-related cellular cytotoxicity.

Sometimes CMV plays a crucial role in the immune response and inflammatory stimulation: it prompts lymphoproliferative disease [7] or a cross-reactive immune response and graft-versus-host reaction [8]. The impact of CMV serologic status can play a significant role in the outcome of hematopoietic stem cell transplantation (HSCT) [9]. Unfortunately, such CMV-IgG serostatus is interpreted as a qualitative parameter only [6,9] and in the context of its influence on final outcome [5,6]. Non-relapse mortality (NRM), and overall or disease-free survival (OS and DFS, respectively) are indirect exponents of CMV infectious process and pathogenicity.

In recent years, there has been renewed interest in the role of bone-marrow-derived NK and B cells in the context of haplo-HSCT and humoral immunosurveillance against CMV. The study sheds light on the host-CMV balance and shows a crucial role of the CMV-specific IgG level. The evolution of an adaptive immune response to CMV has been presented in three different host–virus situations: in immunodeficiency after HSCT, as well as immunocompetent adult or infant with CMV-replication.

## 2. Results

### 2.1. Evolution of CMV Infectious Process in Immunodeficient HSCT Recipient

#### 2.1.1. Humoral CMV Specific Immunity

Recent studies have demonstrated that the complete protection against infection was achieved in animals which lack functional T and B cells by prophylactic or therapeutic administration of polyclonal antibody preparations [10]. Therefore, I examined whether a similar event also occurs in humans, first in a patient after immunoablative conditioning and HSCT from a CMV-negative, HLA-matched sibling donor. To illustrate that such situation is presented in figure (see Conclusions section). The clinical characteristics of this sibling is presented in Table 1. CMV infection was monitored using a real-time quantitative polymerase chain reaction (RT-PCR) in both patient and donor.

In the first month after transplantation a significant decrease of CMV-specific IgG antibodies was observed up to +33 days, when it reached the lowest level, i.e., 93.2 RU/mL. This corresponded with the start of the most intensive CMV replication and most severe lymphopenia (i.e., between 2–11 and 19–01). Furthermore, the increase of CMV-specific IgG as the result of de-novo synthesis with avidity decrease (between 2–11 and 24–11) preceded detection of CMV-specific IgM, as well as significant CMV-quantiferon increase (i.e., after 24–11, Figure 1).

#### 2.1.2. Cellular Immune Response

Since the addition of a T cell depleting agent has been associated with severe lymphopenia and the delay of immune reconstitution, as well as CMV reactivation, I decided to carry out all of the immunoparameter analyses within the first 180 days after HSCT.

Knowing that a low IFN-γ response to phytohemagglutinin (PHA) predicts poor survival [11], I assessed CMV-specific and mitogen-induced IFN-γ release using the same commercial quantiferon-cytomegalovirus assay. Unfortunately, in the quantiferon technique, the interferon immune response after CMV-peptide-laden human leukocyte antigen (HLA) class I stimulation showed higher level than PHA-induced in the first wave of lymphocyte reconstitution (i.e., between 2–11 and 19–01), which of note, was during most intensive CMV reactivation [12].

Previous studies in both solid and stem cell transplant recipients reported that the Quantiferon assay was frequently (up to 38%) “indeterminate” because of a poor mitogen response as a result of immunosuppression [11,13,14]. In this context, for precise interpretation of the quantiferon assay the cytometric analysis, blood count and cyclosporine (CsA) level were performed from the same blood collection (the same venipuncture) and is presented in Figure 1.

A direct analysis of CMV-specific cytotoxic T cell level was performed using a CMV-pentamer technique and compared with the free β2-microglobulin level as a marker of cytotoxic effect [15].

Prolonged lymphopenia and immunoreconstitution with low B and T cell counts (<50 and < 100 cell/µL, respectively) were detected within about three months, i.e., between +1 and +4 after HSCT. It corresponded with weak T cell activity and response to PHA. The flow cytometry with pentamer analysis showed substantial disadvantages. During severe lymphopenia: when CD8-postive T cells were very low with significant difference of blood count, and the flow cytometric analysis showed discrepancy and low reproductivity: within 1 week, the difference was 30–50% (data not shown in Figure 1). Thus, two significant results of lymphocyte population analysis and the pp65-specific pentamer were obtained only after the second wave of immunoreconstitution (as presented in Table 1).

### 2.2. Evolution of the CMV Infectious Process in Immunocompetent HSCT Donor

Due to frequent visits during hospitalization and recipient’s close place of residence, the donor was subjected to routine microbiological control. During the donor’s follow-up visit, mild lymphopenia, lymphadenopathy, and splenomegaly were observed. Neither Epstein–Barr virus (EBV) viremia nor bacterial infection was observed. Stool, urine, and throat cultures were negative. A donor primary CMV infection (CMV-DNA) was detected on day 55 after GCSF mobilization temporally with a low viral load during the peak of CMV-viremia (data not shown). It corresponds with an intermediate β2-microglobulin level and slow affinity maturation. The CMV-specific pentamer analysis was performed (Table 1). The symptoms of CMV mononucleosis were observed until the increase of specific IgG.

### 2.3. Evolution of CMV Infectious Process in Immunocompetent Infant

As prevention of congenital CMV infection via passive transfer of CMV hyperimmune globulin has been reported [16], I tested CMV-IgG level during infancy (Figure 2). Such a clinical model with passively acquired maternal IgG is the second unique clinical constellation (Table 2), when the IgG is the sole factor of CMV-specific immune response [16].

Although the strict moment of infection is difficult to assess and the CMV reactivation is possible (many newborn infants are infected in utero [16]), an inverse correlation of virus replication and the decrease of CMV-specific IgG was observed (Figure 2).

In the first phase the significant reduction of CMV-IgG was observed as a result of the decrease of passively acquired maternal IgG, whereas in the second phase (during CMV-replication process), de-novo produced CMV-IgG was observed, but without IgM. On the contrary, the further spontaneous decrease of maternal EBV-specific IgG, without de-novo synthesis and infection were presented in Figure 2.

## 3. Discussion

Delayed immune reconstitution has also been a major limitation in some haplo-HSCT platforms, especially now with the development of innovative strategies such as cord blood or transplants from haplo-identical family donors. As presented here, despite fast hematologic recovery, prolonged immune reconstitution is still a clinical problem. Interestingly, the level of recovery of thymopoiesis has been directly linked to CMV clearance and survival [18]. 

### 3.1. Humoral Immune Response

#### 3.1.1. CMV—Serostatus

It is a clinical paradigm that anti-CMV-IgG-positive donors facilitate post-transplant immunological recovery and chronic CMV infection affects the immune system [9]. On the other hand, a seronegative donor and conditioning with T cell depleting agents (anti-thymocyte globin or alemtuzumab) are associated with reduced GvHD and the delay in immune reconstitution shows a severe drawback [19]. Weak CD4 increase, preferentially CD8+ expansion and a CD8CD57+ increase were observed (Figure 1). This corresponds with previous data: ratios of CD4+ to CD8+ cells were significantly and persistently lower in the ATG-treated patients within prolonged period (years) with increased regeneration of the CD8+ bright CD57+ subpopulation [19,20]. Of note, following the manufacturer’s (Fresenius) data, ATG shows high reactivity to T (CD2, CD3, CD4/8, TCR), B cells (CD19, CD20, CD40), and plasma cell (CD38) antigens, without reactivity to NK cells. Therefore, the serum IgG level during first months after transplantation reflects the IgG half-life and the decrease of the recipient’s humoral immune response. The lymphocyte B and CD4-helper role was blocked by ATG together with other immunosuppressants (here–CsA, Figure 1). Interestingly, because of a seronegative donor, a residual recipient’s IgG-class antibody was the sole and exclusive factor of CMV-specific immune response during its switching (Figure 1). Contrary to my observation, most of the literature, especially data from EBMT/NMDP database, are devoted to the influence of the donor CMV serostatus on overall survival with conflicting results [5,6,9,21]. Unfortunately, EBMT analysis did not include in vivo T cell depletion therapy [21] and serostatus was analyzed as a qualitative data. In a similar to this case CMV constellation i.e., R+ (recipient CMV-positive), the estimated 5-year overall survival after HLA-identical sibling HSCT was 52% both for CMV-seropositive and -seronegative donors [21].

#### 3.1.2. CMV-IgG Level

It is a clinical paradigm, that IgM/IgG titer increased in sequential samples may indicate active CMV infection [22,23,24]. The prediction with the test results and preemptive therapy was quite difficult when IgM and CMV viremia increased in parallell manner (Figure 1). In my observation, the ganciclovir regimen based on CMV-specific IgM was too late since the antiviral therapy did not prevent further CMV reactivation: a CMV-viremia increase was still observed (Figure 1). Contrary to IgM and the disadvantages of pentamer analysis for CMV-specific IgG levels, CMV-quantiferon analysis and a significant decrease of such specific immunity may be used in clinical practice. An intensive and significant decrease of CMV-quantiferon and specific IgG precedes virus reactivation. During severe lymphopenia, before the second wave of immunoreconstitution, the specific IgG is a “big player.”

The CMV-specific IgG, as the sole factor that prevents virus reactivation, was observed in an animal model: complete protection against infection was achieved in mice which lacked functional T and B cells via prophylactic or therapeutic administration of polyclonal antibody preparations [10]. The same was observed in pregnancy: the only risk factor for an CMV-affected child was the mother not receiving immunoglobulin [16]. Unfortunately, in the case-control study the initial avidity was low and the IgG level was not tested. This communication is the first report of the same preventive role of anti-CMV IgG after HSCT, as well as in infancy. Contrary to numerous publications about primary maternal cytomegalovirus infection, my model shows crucial role of passive acquired high avidity IgG from mother or residual recipient with protective level of such antibody near 100 RU/mL.

Noteworthy, our patient showed a persistently high IgG level, in spite of immunodeficiency, lymphopenia, and virus reactivation observed after the decrease to CMV-IgG = 93.2 or 89.08 RU/mL (Figure 1 and Figure 2 respectively). Within the first weeks of lymphopenia (i.e., between 24–10 and 24–11), the recipient did not develop a CMV infection. The comparable IgG level was observed when CMV viremia developed during infancy, after a significant decrease of maternal IgG (Figure 2). CMV avidity was initially high (residual recipient-derived), but its decrease was difficult in clinical interpretation due to the parallel increase of IgG synthesis using naive B cells from the graft (Table 1). Avidity and the β2-microglobulin level showed good accuracy and reflected differences between primary and secondary infectious processes. A faster increase of avidity (affinity maturation) in the recipient may result from much higher antigenic stimulation in the recipient by CMV-reactivation than in the donor (i.e., primary infection) (Table 1). In the first post HSCT phase, after fast myeloid cell line recovery, IgG may act by follicle (germinal center) formation, opsonisation and complement activation as well as FcγR (CD16,32,64) with several effector antiviral mechanisms as described elsewhere [25].

Of note, a small proportion of T cells was CMV-specific, but NK (defined by CD16+56+ in our analysis) expressed FcγR III (CD16).

### 3.2. Cellular Immune Response

Interestingly, during the crucial CMV reactivation and immune response period, the CMV-induced IFNγ release was higher than that induced by phytohemagglutinin (PHA) (i.e., indeterminate results) because T cells were under the influence of immunosuppressants (ATG and CsA) (Figure 1 and Figure 3). PHA pan-T-cell mitogen and stimulator did not act when severe T-cell lymphopenia was observed and the patient’s blood contained T cell immunosuppressants (here ATG and CsA) (Figure 1). On the contrary, no association was seen between the PHA-induced quantiferon response and ATG regimen [11], but such a statistical analysis was done at 3 months. Likewise, in my observation at 3 months (i.e., 2–11), full chimerism was observed, the PHA-induced quantiferon result was lower than CMV-induced, but a CD16+ 56+ NK cells increase was observed before T cell reconstitution (Figure 1). NK cells, as the first population of lymphocytes, reconstitute in recipients after HSCT [12] and in the CMV replication phase (between 24–11 and 19–01) NK cells are the most abundant lymphocyte population and results for PHA were low. A previous study from haploidentical HSCT showed IFNγ-producing KIR+ and NKG2C+ NK cells that are expanded only in patients with CMV reactivation [12]. NKG2C+ NK cells are transplantable and expand in vivo in response to the recipient CMV antigen [26,27]. It corresponds with my observation: during CMV-viremia (i.e., between 24–11 and 30–01), each tube contained heparinized whole blood with CMV virus during active replication. Therefore, CMV-specific IFN-γ release corresponded with the increase of CD16+ 56+ NK-cells (CD4 and CD8-later) (Figure 1) since Tube 1 (test) contained the patient’s whole blood with NK cells, a CMV-specific IgG with high avidity (Table 1), and various CMV epitopes (both natural and synthetic). Contrary to IFN-γ, the CMV-induced IL-2 release by T cell yielded inferior and insignificant results compared to IFN-γ [28]. These results support a greater role of NK cells than that of T cells but this hypothesis needs further studies for confirmation. The second question is whether such a reaction of NK cells is IgG-dependent. Such epitope–paratope binding was described in details only for CMV glycoprotein B (gB), underrepresented in quantiferon technique (see Appendix A with table). Noteworthy, for CMV only two residues Y364 and K379 form a discontinuous epitope and they were identified as essential for CMV and IgG binding [29]. A quantiferon-CMV assay strongly correlated with CMV-specific antibodies [30]. Contrary to the IgG level in my observation and to IgG serostatus before HSCT as described previously in my center [9], there was no difference in the quantiferon test between participants who subsequently developed CMV disease, CMV reactivation, or spontaneous viral control [31].

### 3.3. Innate Immune Response

The short time of IgG switching (between 24–10 and 24–11) must be seen against the innate immunity background. Latest findings indicate that viral dsRNA may initiate fast immunoglobulin class switching, frontline IgG and IgA responses through an innate TLR3-dependent pathway without Th/B cooperation in follicular niche [17]. CMV is a strong TLR3 ligand [32], therefore the fast immunoglobulin class switching and serological evolution observed here, i.e., increase of IgG before CD4 reconstitution, may be TLR3-dependent (Figure 1). Furthermore, in the interferon gamma release assay, TLR3 agonists polyinosinic-polycytidylic acid (Poly(I:C) augmented IFN-γ responses after a challenge with recombinant and synthetic CMV pp65 peptide, as described elsewhere [28]. Interestingly, in the CMV-immune response, the TLR agonists constitute a costimulating reagent for IFN-γ [17,28], but unfortunately with negligible effect on IL-2 release by T cells [28]. Paradoxically, the TLR3-mediated signal during CMV viremia may be the source of “natural” quantiferon optimization as well as the crucial factor in host-CMV balance.

## 4. Material and Methods

### 4.1. Material

The HLA-matched sibling donor was CMV-specific IgG negative (D−). However, the recipient was a CMV-positive 58-year old woman with acute myeloblastic leukemia (AML) who received hematopoietic stem cell transplantation after the first complete remission. She received reduced-intensity and immunoablative conditioning with Busulfan-Fludarabine and anti-thymocyte globulin (Bu-Flu-ATG [32]) i.e., 8 mg/kg BU, 180 mg/m^2^ FLU, and 40 mg/kg ATG 30 (Fresenius).

Graft-versus-host disease (GvHD) prophylaxis consisted of CsA alone. Mild aGvHD was observed. Because of CMV reactivation, lymphopenia, and ATG regimen, the patient was subjected to a narrow immunosuppressive regimen with a higher CsA dose up to the 250 µg/mL level. Engraftment of hematopoietic stem cells and hematologic recovery were observed within the second week: neutrophil count over the first 14 days of the treatment was below 1000/µL and red blood cells with donor blood group antigens were found at +25 days. Chimerism in myeloid-derived populations of polymorphonuclear cells (i.e., granulocytes) was achieved within the first months. Full chimerism was observed in the second month after HSCT in the marrow, whole blood, as well as lymphocyte population (Figure 1). The patient did not receive an immunoglobulin or serum transfusion. 

Details of demographic and clinical parameters of the donor and the recipient are summarized in Table 1.

### 4.2. Methods

Several available methods are used for monitoring host-CMV balance. Therefore, the evolution of adaptive immune response to CMV was observed during immune system switching using CMV serology (humoral immune response) with enzyme-linked immunosorbent assay (ELISA), and CMV-quantiFERON and pentamer analysis (cellular immune response), as well as CMV-viremia with real-time quantitative polymerase chain reaction (RT-PCR).

Microplate ELISA (Euroimmun product EL 2570-9601 G) was used with native CMV antigens and linear calibration 2/20/200 RU/mL VIDAS CMV IgG 30204 [28]. The assays were mostly based on viral lysate and antigens derived from CMV strain AD169. Such cell lysate of cytomegalovirus gave a wide spectrum of viral proteins. Sera were incubated in microtiter plate wells. Specific anti-CMV-IgG antibodies, and if attached to the coated wells, were identified with peroxidase-labeled anti-human IgG antibodies [9,28]. The intensity of the color produced was proportional to the concentration of antibodies in the serum sample. For avidity, the Euroimmun ELISA test was applied with avidity buffer 1 containing urea (product ZF 1131-0101-1). An avidity index (AI), was calculated using the formula AI = (result of the well with avidity buffer/ of the well with regular buffer) × 100 (expressed as a percentage). Cut-off low- and high-avidity scores were <40 and >60, respectively.

Qualitative CMV DNA analysis in peripheral blood cells was determined using real-time PCR with Light Cycler II (Roche, Mannheim, Germany) and expressed in numbers of CMV copies as described previously [9].

Peptides representing HLA-A × 0201–restricted viral epitopes (i.e., tegument protein pp65) were used in pentamer analysis as described elsewhere [9]. All data were acquired from FACSCalibur 2 (Becton Dickinson, San Jose, CA, USA). Flow cytometry analysis and evaluation of B/T/NK cells population were carried out in accordance with the guidelines in the EBMT Handbook 2012 Edition.

#### Interferon Gamma IFNγ Release Assay (IGRA)

IFNγ release under the influence of CMV epitopic peptides quantiFERON-CMV assay (QIAGEN) (named here as quantiferon) was used for cellular immune response monitoring [11,21,31]. According to the manufacturer’s details, quantiferon is an in vitro assay using a peptide cocktail to stimulate cells in heparinized whole blood (cat. No 0197-0201). Noteworthy, most of the peptides used in this technique are derived from the CMV pp65 antigen (i.e., 15 out of the 22) (Appendix A). The HLA specificities are also distributed unevenly (e.g., HLA-A24 -three peptides, A1-only one). Each set contained three tubes:(1)CMV-peptide-laden human leukocyte antigen (pHLA) class I, presented in Appendix A.(2)Tube 2 contained phytohemagglutinin (PHA), i.e., pan-T cell mitogen as a positive control.(3)Tube 3 contained sterile phosphate-buffered as a negative control.

After blood collection, the tubes were gently shaken and then incubated overnight for 18–24 h at 37 °C. All of the tubes were then centrifuged and plasma was collected. Detection of interferon gamma (IFN-γ) was done using an enzyme-linked immunosorbent assay (ELISA) (QIAGEN cat No.0350-0201 QuantiFERON ELISA). Noteworthy, contrary to the ELISPOT technique, the quantiferon-CMV test evaluates the IFN-γ production in a volume of 1 mL of whole blood, and may be modified by severe lymphopenia and immunosuppressants, while the ELISPOT test considers the IFN-γ production in a given number of peripheral blood mononuclear cells (PBMCs) isolated from blood [28,30].

## 5. Conclusions

Immunological escape after HSCT (Figure 1) and the decrease of maternal IgG after birth (Figure 2) show a good model for assessing the protective IgG level against CMV reactivation. These are unique situations with different populations at risk (Table 2).

Marrow-derived NK cell predominance was observed in the early phase of immune reconstitution after ATG conditioning (Figure 1). It may be the first line antiviral immunity after HSCT [12], especially with immunoablative conditioning with ATG (Figure 3). The presence of several confounding factors, and very differential, sometimes impenetrable, immunological parameters in dynamic clinical situation that cause strict immunological circumstances, for example presented here, get out of control and simple classifications. Therefore, statistical analysis is misleading and biased [5,6,9,21]: these calculations contain hundreds of factors, but one of them is specific IgG level before transplantation, since it is the one and exclusive specific element of immune response within 2–4 months. CMV-IgG serostatus is not a qualitative but dynamic and quantitative feature. Apart from the virus-specific cellular immunity, the humoral arm of the immune response plays an important role in the containment of CMV infection.

The recipient decrease of CMV-specific IgG level and corresponding CMV quantiferon release were better for clinical decision than IgM and HLA-restricted CMV pentamer analysis. Indeterminate results (from low mitogen values) of simple and universal QuantiFERON-CMV should be interpreted against clinical and immunoparameter background (i.e., virologic, innate, serological, blood count, and lymphocyte population analysis). CMV-induced innate immune response, secretion of IFNγ via NK cells require further prospective research in the future. Furthermore, data collected supports the possible effectiveness of hyperimmunoglobulin infusions for the prevention of congenital or posttransplant CMV disease.

## Figures and Tables

**Figure 1 ijms-20-00271-f001:**
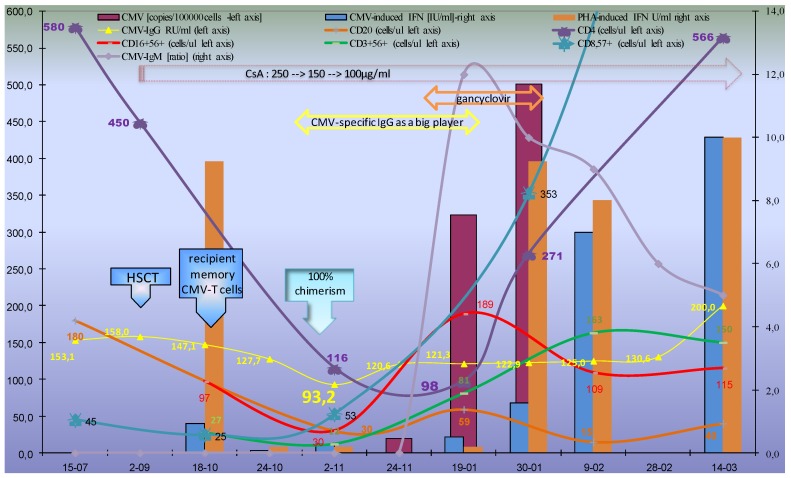
Host-CMV balance: decrease of CMV-specific immunity and subsequent reconstitution during lymphocyte recovery after hematopoietic stem cell transplantation (HSCT). CMV-specific immune reconstitution during lymphocyte recovery after HSCT was tested using CMV-IgG and –IgM, as well as quantiferon analysis (interferon gamma release in whole blood under the influence of CMV-epitope cocktail as described in the Appendix A). The diagram shows the timeline with the results presented on the vertical axes: CMV-specific IgM, CMV-and PHA-induced IFNγ release (right vertical axis), as well as CMV-IgG, cell, and CMV counts (left vertical axis). For a broader and strict interpretation of quantiferon assay, a therapeutic regimen and cytometric analysis of the major lymphocytic populations were presented. The short time of IgG switching (between 24–10 and 24–11) must be seen against the backdrop of long-term cellular switching between 18–10 and 9–02) during the CMV reactivation process. Noteworthy, during the most severe lymphopenia (i.e., between 2-11 and 19–01) the sole factor of CMV-specific immune response was a residual recipient CMV-IgG with high avidity and newly produced donor-derived IgG later (with decreased avidity after full chimerism). The quantiferon technique during CMV-viremia contained CMV-infected cells: in the presented case, 20,323 and 501 copies per 100,000 cells (purple columns for 24–11, 19–01 and 30–01, respectively). This corresponded with the increase of NK cell count, as the first population. Furthermore, presence of an additional “professional” antigen presenting cells in the whole blood may improve the activation threshold for cytokine synthesis via virus-specific T cells resulting in enhanced IFN-γ responses.

**Figure 2 ijms-20-00271-f002:**
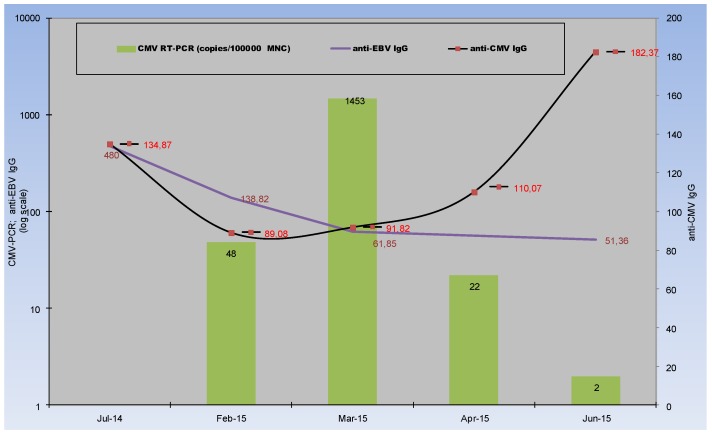
CMV-specific evolution during infancy. The diagram shows the timeline with the results presented on the vertical axes anti-CMV IgG (right) as well as anti-EBV IgG and viremia (left vertical axis). Mild and unspecific symptoms of viremia were observed when the passive-acquired maternal IgG disappeared. Increase of specific IgG corresponds with virus clearance. For comparison spontaneous decrease of maternal EBV-specific IgG was presented without infection and de-novo synthesis (the continuous decrease of EBV-IgG was observed). Significant decrease of maternal CMV-specific IgG precedes virus replication.

**Figure 3 ijms-20-00271-f003:**
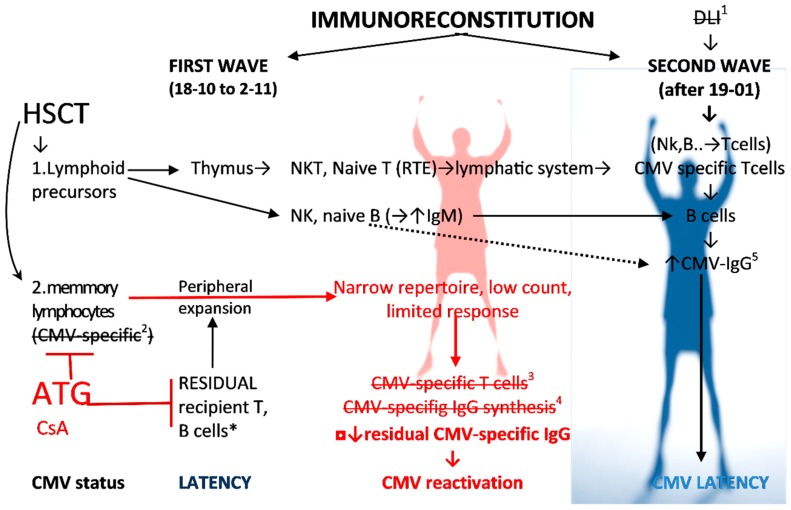
CMV specific reconstitution as a hallmark of adoptive immunorecovery after HSCT. The figure shows the evolution of an adoptive immune response to CMV against the backdrop of the influence of clinical factors in the presented case (Figure 1) with ineffective first wave immunoreconstitution. The crossed-out elements are absent in the presented case: ^1^ DLI as well as ^2^ donor memory (and effector) lymphocytes from the graft in our case do not contain CMV-specific cells (donor was seronegative D−) (see Table 2). * Residual CMV-specific recipient T and B cells are blocked by the lymphocyte depleting conditioning. The T cell compartment was also blocked using immunosuppressive therapy with CsA. ^3,4^ ATG blocks the first wave of renewal in a significant way. It works on cellular compartment (i.e., cells by CD2, CD3, CD4/8, TCR) as well as humoral counterpart (B cells by CD19, CD20, CD40, and plasma cells CD38), and therefore CMV-IgG synthesis. Lymphocyte repertoire is in first wave very narrow because of lymphocyte depletion. Small amount of NK, B lymphocytes and (to a lesser extent) T/NKT cells survived the conditioning. Therefore, in my report the first wave of adoptive immunoreconstitution was weak (without CMV-specific T and B cells, and CMV-IgG synthesis). Noteworthy, for months, the sole and exclusive factor of CMV-specific immune response were the recipient’s residual IgG with their gradual decrease (Figure 1). ^5^ De novo IgG synthesis may be observed without T cells and IgM switching as the result of CMV and TLR3 interaction [17] as indicated by the dotted line. Abbreviation: RTE—recent thymic emigrants, ATG—anti-thymocyte globin, CsA—cyclosporine A.

**Table 1 ijms-20-00271-t001:** Recipient and donor demographic data and crucial immunoparameters.

Feature	Recipient (−14 Day)		Donor
Age	56 y		64y
Gender	Female		Male
Blood group	A Rh-positive		B Rh-positive
Serostatus (IgM)	CMV, EBV, VZV, HSV-negative		EBV, HSV-negativeCMV (−) →(+) ^5^
Serostatus (IgG)	CMV, EBV, HSV-positive		CMV 0,12 (−) →63.05 (+) ^5^ HSV, EBV -positive
Body weight (kg)	58		86
Lowest WBC	250 (+10 day)		2800
Lowest lymphocyte count	190 (+31 day)		650
CMV-IgG AWIDITY ^1^ (%)	High (till +60 day) Intermediate (60–84 day) Low (84–150 day) High (>180 day)		Low (till + 115 day) Intermediate ^6^
**Flow Cytometry ^2^**	**19–01**	**30–01**	**During CMV viremia**
CD4+CD27+CCR7+CD45RA+ Naive	0.05	0.23	NT
CD4+CD27+CCR7+CD45RA- Central memory	0.69	2.22	NT
CD4+CD27-CCR7-CD45RA- Effector memory	5.30	4.16	NT
CD4+CD27-CCR7-CD45RA+ CD45RA + effector memory T cells (TEMRA)	0.41	0.15	NT
**Pentamer Analysis ^3,4^ (% of CD8 lymphocytes)**			
CD8^high^ + CMV pp65+	2.32	6.91	0.08
CD8^high^+CD57 + CMV pp65+	0.9	2.32	0.00
β2-microoglobulin^4^ (mg/L)	6.4	9.2	3.0

^1^ Avidity index (AI), calculated using the formula AI = (OD of the urea-washed well/OD of the well washed with regular buffer) × 100, expressed as a percentage. The AI cutoff point for defining low avidity is <40%, whereas the AI cutoff point for defining intermediate and high avidity is 40–60% and >60% respectively as described in Euroimmun test. ^2^ Flow cytometry analysis and evaluation of naïve and memory T cells population were carried out in accordance with the guidelines in the EBMT Handbook 2012 Edition. Human T cells can be divided into at least four different subsets, based on their phenotype and functions: naive (T naive, CCR7 + CD45RA+) and central memory (TCM, CCR7+ CD45RA−) T cells display high proliferative potential and lack of an immediate effector functions whereas effector memory (TEM, CCR7−CD45RA−) and CD45RA+ effector memory (TEMRA, CCR7−CD45RA+) T cells have low proliferative capacities but produce cytokines and exert cytotoxic activity, respectively. Data were expressed as the % of lymphocytes. ^3^ Peptides representing the HLA-A × 0201–restricted viral epitopes, which is derived from the CMV tegument protein (pp65) was used in the pentamer technique: before the reconstitution and the +110 days after a significant HSCT decrease below the detection limit (0.01%) was observed or with high discrepancy in the pentamer analysis. ^4^ Contrary to avidity, the β2-microglobulin level showed good accuracy and reflected differences between primary and secondary infectious process. The good correlation of β2-microglobulin and viremia level was observed. ^5^ Serological analysis of CMV was shown for the time just at date of Granulocyte-colony stimulating factor (G-CSF) start (in peripheral blood stem cell mobilization) as well as +47 and +55 for IgM and IgG respectively. ^6^ IgG avidity was low till +115 and intermediate from +125 day after start of G-CSF (granulocyte colony-stimulating factor) In contrast to the recipient no high avidity was observed up to +200 days.

**Table 2 ijms-20-00271-t002:** The unique host–virus constellations and populations at risk in transplantology and obstetrics.

	Immune Response	Primary	Secondary (Days)
Infection			
Primary (by mucosa, skin)		Typical IgM→IgG	DLI ^2^, Breastfeeding MFT
Secondary (i.e., reactivation *)		HSCT ^1^, neonatal CMV (without IgM **) solid organ transplantation	Typical (IgG only)

The unique host–virus constellation after hematopoietic stem cell transplantation (HSCT, Figure 1) or in neonatal CMV infection was presented (Figure 2) contrary to typical constellation, i.e., primary immune response during primary infection (Table 1, i.e., in immunocompetent donor). ^1^ In HSCT with D-R+ constellation: increase of specific IgG (within 1–2 weeks) shows primary immune response manner (but ** sometimes without detectable IgM increase) after virus reactivation (endogenous infection, not by portal of entry). Under the effect of viral particles fast, TLR-mediated class switching occurs without Th and CD40L as described in discussion. On the contrary, such a situation was observed in the D+/R- constellation in solid organ (especially liver, pulmonary) transplantation. * Noteworthy: the primary immune response was induced in lymphoid tissue in many places and organs at the same time; therefore, the antigenic stimulation was more intense than in a physiological manner (route of transmission → the portal of entry: the site of infection is localized). It corresponds with a much higher β2-microglobulin level in the recipient than in the donor and faster (about +180 day) increase of avidity (affinity maturation) (Table 1). ** Because of fast TLR3-mediated immunoglobulin class switching [17], the IgM may be absent (Figure 2) or after the IgG increase (Figure 1). ^2^ In D+/R-after donor lymphocyte infusion (DLI) that contains CMV-specific T and B lymphocytes, the secondary immune response was observed after primary infection (D+/R- constellation). The same phenomenon was observed in neonates after MFT (maternal-fetal transplantation), breastfeeding (milk contains 10–20% lymphocytes), and solid organ transplantation.

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
