# Peer review of "CMV-Specific Immune Response—New Patients, New Insight: Central Role of Specific IgG during Infancy and Long-Lasting Immune Deficiency after Allogenic Stem Cell Transplantation"

_ijms, 2019, doi:10.3390/ijms20020271_

Round 1

Reviewer 2 Report

The author has addressed my major concerns such that the science presented in the manuscript is suitable for publication.  However, I still feel that employing an editor for English language use would improve the presentation of the paper and increase its readability, especially for a non-expert audience.

Author Response

Answer: Done.

The relevant explanations are added in the figures and text (see answer for Reviewer 2) as well as clarification with a native English speaker.

Reviewer 3 Report

Scientifically this story is interesting as it provides new observation about the virus-host interaction from different kind of patient samples. Therefore, I think this is consider novel and readers should pay attention to this new observation. In addition, I was satisfied with the experimental details provided in this revised version.

Authors need to include more details in the figure legend (for example, Figure 1) in order to help readers to understand the results.

Figure 1 x-axis and y-axis information are missing.

Author Response

Issue 1:

Authors need to include more details in the figure legend (for example, Figure 1) in order to help readers to understand the results.

Answer: Done

To understand without reading the main text of the manuscript the abbreviations, informative details have been added into the figure description.

ISSUE2: Figure 1 x-axis and y-axis information are missing.

Figure 1 and 2 descriptions contain data indicated by: on left (y-axis) against the right-hand scale

The diagrams show the timeline (vertical line).

This manuscript is a resubmission of an earlier submission. The following is a list of the peer review reports and author responses from that submission.

Round 1

Reviewer 1 Report

This is not written properly. I believe it is based on a single case - but the basis of selection of this patient is unclear. For example. Fig 1 has no scale for most parameters and is a mixture of data and supposition. Table 1 and Fig 1 could perhaps be exchanged for a panel of individual graphs aligned by sample date.

I would also recommend that the manuscript be reduced in length by at least 50% with omission of factors not relevant to the key argument (T-cells vs Ab).

Note you can assume that your readers have a thorough knowledge of immunology.

I cant understand Fig 2. Why are some items crossed out?

Author Response

1) Question/issue1
This is not written properly. I believe it is based on a single case - but the basis of selection of this patient is unclear.

Answer:

Main concept and idea is presented in Table 2 and Figure 3 as well as graphical abstract (simplified form).

a) The patient selection was based on unique donor/recipient constellation (D-/R+) to observe CMV-specific immune response switching.

b) In addition, in the presented case the decrease (and finally disappearance) of secondary cellular immune response was observed after conditioning with ATG.
Therefore, the primary immune response is not in physiological manner (during multiorgan reactivation and severe CMV-viremia). The role of humoral (class IgG) immune response was observed when the T cell level was very low, the CMV-induced interferon release was not observed (between 24-10 and 2-11) without CMV reactivation. Subsequent decrease of IgG (below the “protective level”) caused CMV reactivation: virus replication exceeded the host CMV-specific immune response. On the contrary, CMV (-) haploidentical family donor was a good example for the observation of primary infection in immunocompetent host.

2) Q/issue 2

For example. Fig 1 has no scale for most parameters and is a mixture of data and supposition. Table 1 and Fig 1 could perhaps be exchanged for a panel of individual graphs  aligned by sample date.

Answer: Done.

a) Indeed –host-virus interaction is a multi-faceted phenomenon. It calls for an integrated response involving various titers of cellular (CD4 and CD8 T cells), humoral (IgG) and innate (Nk, interferon) immunity. The cytometric analysis is a part of interpretation of QuantiFERON (see later)

i) To show crucial IgG role I had to show other elements, their decrease and finally disappearance.

ii) The short time of IgG switching (between 24-10 and 24-11) must be seen against this background (long-term cellular switching between 18-10 and 9-02).
The sentence was added tofigure description below the figure.

b) Figure 1 has been changed.
In figure legend above plot the units and corresponding Y-axis are described.If we look at the Y-axis of this graph, we will see that most parameters are presented on the left Y-axis. The CMV-quantiferon values and IgM level are shown on the right axis.

c) Simplified form of Figure 1 is presented in graphical abstract: the CMV-specific immune response switching from recipient to donor-derived is presented.

3) I would also recommend that the manuscript be reduced in length by at least 50% with omission of factors not relevant to the key argument (T-cells vs Ab).
ANSWER: DONE

4) Note you can assume that your readers have a thorough knowledge of immunology

ANSWER

a) DONE. Thank you for the suggestion. The descriptions of IgG activity, mechanisms of action are unnecessary and may impair the clear message and novelty of the observation. 

5) I can’t understand Fig 2. Why are some items crossed out?

ANSWER

a) Figure 2 shows CMV-specific immune response against the background of the typical immunoreconstitution after HSCT:

i) First wave -from memory T cells from recipient or graft

ii) Second wave -from naive –donor-derived lymphocytes

b) Sometimes, as presented in this case for CMV, the first wave is not observed or insignificant (Figure 1). For clarification a sentence was added: The crossed out elements are absent in the presented case: 1 DLI as well as 2donor memory (and effector) lymphocytes from the graft in our case do not contain CMV-specific cells (donor was seronegative D-)

Reviewer 2 Report

The case report submitted by Dr. Zdziarski describes peripheral blood stem cell transplantation from a haploidentical cytomegalovirus-(CMV)-seronegative female sibling to a CMV-seropositive recipient. The case is well monitored for laboratory immunological features as well as virological and clinical features, which are described in detail in the text, tables and figures. As a result of the monitoring, the author reaches the conclusion that CMV-specific immunoglobulin levels are critical for control of CMV reactivation and played a major role in the virological history of this case. It is a very comprehensive case report from which the author relates many aspects towards a general review of features involved in determining outcomes related to CMV infection or reactivation in immunosuppressed individuals. The author provides a thorough description and analysis of events relevant to the case which can be insightful and provocative.  However, there are several major and minor issues with the manuscript as presented that I feel reduce its potential as an informative and impactful case report.

1. The English language usage needs significant improvement. While the text is generally understandable, there are numerous cases where it’s not clear to me if the author is actually making a provocative statement or if it is just the English. For example, is CMV infection the most common viral infection or just very common.  There are too many instances where small changes in wording would clarify the text considerably.  Caveats of the author’s conclusions and speculation should also be included where appropriate.

2.  Some of the technical methods need additional description.  What is the quantiferon kit measuring?  Can NK cells and anti-CMV Ig be responsible for the IFN production?  How?

3.  How ablative is the ATG treatment.  Are all recipient T cells eliminated or is there some residual survival?   Can it be certain that there is no lymphocyte chimerism in this case?  It is an assumption that CMV might be activating de novo B cell class switching through TLR3 in the absence of CD4+ T cell help.  The figures included contain lots of information, which makes them complex and difficult to grasp.

4.  An oligoclonal response of CMV-specific CD8+ T cells post transplant has been reported. Could this possibly explain the greater CMV-specific than PHA-driven IFN production, rather than NK cells? CD8+ T cells appear to increase before CD4+ T cells and around the same time as the NK cell increase.

5.  What does the fall to 93 IU actually indicate about anti-CMV Ig levels?  Is this necessarily a non-protective level?

6.  The discussion should be shortened considerably to focus on the case itself and issues from the case that are generalizable for the bigger picture in some novel way

Author Response

QUESTION/ISSUE

1) The English language usage needs significant improvement.  While the text is generally understandable, there are numerous cases where it’s not clear to me if the author is actually making a provocative statement or if it is just the English. For example, is CMV infection the most common viral infection or just very common.  There are too many instances where small changes in wording would clarify the text considerably.  Caveats of the author’s conclusions and speculation should also be included where appropriate.

ANSWER: DONE.

The too long, ambiguous and doubtful sentence was modified/ canceled. When the conclusion or sentence was hard to read explanations with citing of literature have been added.

2) Some of the technical methods need additional description.  What is the quantiferon kit measuring?  Can NK cells and anti-CMV Ig be responsible for the IFN production?  How?

Answer

Thank you for asking. For the strict and exact description of my observation I have to modify the article structure (as Short Communications, expand the Material and methods)
When we look at the simple procedure the interpretation of results is sometimes difficult without laboratory details. The patient’s history, immunostatus is important in the appropriate use of test and interpretation. 

a) In detail (official data of manufacturer):

i) QuantiFERON-CMV is an in vitro diagnostic test using a peptide cocktail simulating human cytomegalovirus proteins (CMV) to stimulate cells in heparinized whole blood. The universal and simple method may be the source of much information, crucial for patients’ health, but strict

ii) Detection of interferon-gamma (IFN-γ) by Enzyme-Linked Immunosorbent Assay (ELISA) is used to identify in vitro responses to these peptide antigens that are associated with CMV infection. Loss of this immune function may be associated with development of CMV disease. The intended use of QuantiFERON-CMV is to monitor the level of anti-CMV immunity in persons at risk of developing CMV disease.

iii) Peripheral blood was collected in three (3 × 1 ml) heparinized tubes.

(1) Tube 1 contained a mixture of human cytomegalovirus peptide epitopes (contrary to pentamer –not HLA-restricted).  The CMV peptides are designed to target cytotoxic cells, including A1, A2, A3, A11, A23, A24, A26, B7, B8, B27, B35, B40, B41, B44, B51, B52, B57, B58, B60 and Cw6 (A30, B13) HLA Class I haplotypes, covering >98% of the human population. Such haplotypes contain various CMV-epitopes.
Contrary to our patient during severe T cell lymphopenia, healthy individuals CMV+  have CD8+ T cells in their blood that recognize these antigens or PHA.

(2) Tube 2 contained phytohemagglutinin (positive control) –  pan-T-cell mitogen

(3) Tube 3 contained sterile phosphate-buffered saline (negative control).

After blood collection by venipuncture, the tubes were gently shaken (10 times up and down) and then incubated overnight for 18–24 h at 37 °C. All of the tubes were then centrifuged and plasma was collected. Specific IFN-γ levels were measured using a standard enzyme-linked immunosorbent assay.
According to the manufacturer, the results were considered positive when the peptide response was ≥0.2 IU/ml of IFN-γ.

 FAQs and Troubleshooting on manufacturer’s web page show ambiguous explanation of low PHA values: Indeterminate results (from low Mitogen values) would not be expected to change on repeat unless there was an error with the ELISA testing

b) IFNγ is produced predominantly by natural killer (NK) and natural killer T (NKT) cells as part of the innate immune response, or by CD4 Th1 and CD8 cytotoxic T cells in specific manner.
When we look at patient’s clinical and laboratory data after HSCT Blood count dropped, the severe lymphopenia is observed. Diagnostic test selection depends largely on the clinical purpose.
The cytometric, immunoserostatus analysis and blood count are a part of interpretation of QuantiFERON
CMV results. Therefore, cumulative data are shown in Figure 1.

It is intellectual value of the observation: pan-T-cell mitogen and stimulator does not act when severe T-cell lymphopenia is observed as well as patient’s blood contains T cell immunosuppressant.  The immunosuppressive agents (ATG, CsA) do not modify Nk, that are first wave immunoreconstitution. The observation corresponds with another publication that NK cells, as the first population of lymphocytes that reconstitute in recipients after HSCT (e.g. Oncotarget, 2017, Vol. 8, (No. 1), pp: 51-63). The publication shows that IFNγ-producing NK cells expand in haplo-HSCT patients with CMV reactivation as a result of significant KIR and NKG2C expression. NKG2C+ cell expansion may occur independently of the T-cell receptor (TcR) specificity. ( Imprint of human cytomegalovirus infection on the NK cell receptor repertoire Gumáet al Blood 2004 104:3664-3671.)
Of note, in humans, the killer cell immunoglobulin (Ig)-related receptors (KIRs) consist of either two (KIR2D) or three (KIR3D) Ig-like domains and recognize polymorphic HLA in similar manner as to immunoglobulin (Nature volume 405, pages 537–543)),

c) The peptide cocktail may induce Nk cell response and IFN secretion by two pathways:

i) In our case indirect pathway probable:  CMV-specific IgG was positive, Nk cells were defined as CD16+56+, therefore expressed FcγR III (CD16).

ii) It is highly likely that direct stimulation of Nk cells by CMV-peptide cocktail is in the assay second pathway i.e. by activatory Killer Immunoglobulin-like receptors. Members of the killer cell immunoglobulin-like receptor (KIR) family recognize particular peptide-laden human leukocyte antigen (pHLA) class I molecules. Such CMV-peptide laden HLA is contained in tube 1.

These data do not entirely exclude that the IFN might be preferentially coexpressed by CD8 subsets specific for viral antigens (Since 30-01 i.e. significant contribution of CD8 in lymphocyte population)  but strongly suggest that the IFN release is T-cell independent, since all CD4 and CD8 T cells activated by mitogen (PHA) produce twice lower IFN count.

Noteworthy,activating receptors have lower affinity for their ligands than inhibitory receptors Raulet DH, Vance RE, McMahon CW. "Regulation of the natural killer cell receptor repertoire". Annual Review of Immunology.2001 19: 291–330.

3. (a)How ablative is the ATG treatment.  (b)Are all recipient T cells eliminated or is there some residual survival?   Can it be certain that there is no lymphocyte chimerism in this case?  It is an assumption that CMV might be activating de novo B cell class switching through TLR3 in the absence of CD4+ T cell help.  The figures included contain lots of information, which makes them complex and difficult to grasp.

ANSWER

a) The ATG effect is very pleiotropic: little is known about long-term changes induced by ATG. Müller et al showed in the cited publication that five years after transplantation CD4+ to CD8+ cell ratio of x=0.6 with  a persistent depletion of the CD4+ cells and an increased regeneration of CD8+ cells, in particular of the CD8+bright CD57+ subpopulation.

i) The very low CMV-specific CD8 count was observed probably as a result of immunosuppressive regimen, ATG. In addition the graft does not contain CMV-specific T cell since the donor was CMV(-) Even during viremia CMV- specific T cells  accounted for only 0,08% of all T cells. It was on day 55 after GCSF mobilization. (see table 1).

b) The narrow, restricted repertoire of T cells in first wave of immunoreconstitution is a well described phenomenon. Some recipient lymphocyte survival was observed (till full chimerism) as presented in figure 1; we observed a positive CMV-quantiferon as the result of recipient memory rather. On the contrary, the recipient lymphocyte disappeared after November-2.

In the CMV context the first wave does not contain CMV-specific CD8 T cells: pentamer analysis is quite the equivocal result with significant discrepancy.

On the contrary, PHA –pan-T cell mitogen gives polyclonal, wide-range (CD4 as well as CD8) activation.  If the entire population of T cells secretes such a small amount of IFNγ, a small portion of it (difficult to quantity in pentamer technique) cannot produce more, as much as observed between November and January.

2) An oligoclonal response of CMV-specific CD8+ T cells post transplant has been reported.  Could this possibly explain the greater CMV-specific than PHA-driven IFN production, rather than NK cells?  CD8+ T cells appear to increase before CD4+ T cells and around the same time as the NK cell increase

a) ANSWER

High contribution (about 10%) of CMV-specific CD8 T cells after HSCT in cited article (32). Unfortunately, the patients in the study had a non-immunoablative conditioning. The CMV-specific T cells are not PHA-resistant; together with other T cells produce IFN under the influence of PHA. Of note, most of CD8 T cells express CD57, thus show low proliferative potential.  Noteworthy: the small proportion of T cells was CMV-specific, but most of Nk express KIR and FcγR III (CD16). Some of Nk express inhibitory KIR, therefore do not induce cytotoxic effect on target cells, but secrete IFNγ as described.   EBMT analysis did not include in vivo T cell depletion therapy.

b) significant CD8+CD57+ increase after full chimerism (also in lymphocyte population)  is not translated into CMV-specific IFN release (Fig.1) and CD57+ lymphocytes consist of about one third of CMV-pp65+CD8+ cells (table 1)

c) Interestingly, frequencies of CMV-specific CD8(+) T cells were significantly higher in immunosuppressed individuals than in healthy donors. (ref 28 -Gamadia et al Blood. 2001 Aug 1;98(3):754-61.)

5. What does the fall to 93 IU actually indicate about anti-CMV Ig levels?  Is this necessarily a non-protective level?

Answer:

Yes, the decrease to 93 IU indicates the start of CMV reactivation during severe lymphopenia. Of note, the decrease precedes the negative result of CMV-quantiferon and viremia (November 24). IgG was the only specific element (with stable level). When the CMV-IgG level falls below 100 IU during severe cellular immunodeficiency, reactivation develops. To support the concept, simple presentation of the decrease of maternal IgG and CMV viremia during infancy was introduced.

6.  The discussion should be shortened considerably to focus on the case itself and issues from the case that are generalizable for the bigger picture in some novel way

Answer:

The work was focused on two general issues: 1) observable and underestimated role of CMV-IgG and 2) potential role of Nk
The direct recognition of virally-induced ligands by activating NK cell receptors has been observed in mice (Arase et al. 2002), but is yet to be confirmed in the human system. Together, these studies suggest that activating KIR recognize altered forms of HLA class I that may be up-regulated during CMV viral infection

Round 2

Reviewer 1 Report

The revison has improved this work but it is still lacks the precision of writing required for publication

For example.

The First line of the Results refers to "the recipient".this is described later in the M&M, but the paper must make sense when read in sequence.

Figure 2 refers to levels of IgG during infnacy...from what samples?

It isnt clear which assessment of CMV IgG is presented

Many abbreviations are not defined before their first use

Line 201....CMV serostatus does not indicate a "lack of latency"...abundant literature would show this.

Reviewer 2 Report

The author has addressed the issues I raised and included a second case that supports the assertion that anti-CMV IgG plays an important role in controlling viral reactivation.  This has improved the manuscript and there is important data and information conveyed, but I still feel that the manuscript requires improvement for its appreciation by a larger audience.  The two primary issues are that it still requires editing for English language usage and the discussion remains too long and unfocused. In addition, there is some data presented that I am unable to grasp the significance of and some explanations/interpretations remain highly speculative.

There needs to be more explanation of the pp65+ data included for CD8+ T cells in Table 1. It’s not clear what the numbers represent, what time points they correspond to or even whether they refer to the donor or recipient response.  It’s also not clear what is happening with the IgG and IgM anti-CMV.  Are there recipient memory B cells remaining that may be making anti-CMV IgG before the donor-derived B cells begin to make anti-CMV IgM?

I’m not convinced by the argument that the NK cells are being stimulated by either the residual anti-CMV IgG binding to peptides in the quantiferon kit (this shouldn’t activate NK cells through CD16) or by KIR binding to HLA/peptide complexes.  Some precedent for this is may come from NKG2C binding to HLA-E bound CMV peptides, but this is also unclear.  Is the interferon production sufficient at this point to assume anything is responding much above background?

There are typographical errors on figure 1 (cellls and memory).